

# The global $O_2$ airglow field as seen by the MATS satellite: strong equatorial maximum and planetary wave influence

Björn Linder[1], Lukas Krasauskas[1], Linda Megner[1], and Donal P. Murtagh[2]

[1]Department of Meteorology, Stockholm University, Stockholm, Sweden
[2]Department of Space, Earth and Environment, Chalmers University of Technology, Gothenburg, Sweden

**Correspondence:** Björn Linder (bjorn.linder@misu.su.se)

**Abstract.** The Mesospheric Airglow/Aerosol Tomography and Spectroscopy (MATS) satellite was launched in November 2022, carrying as its main instrument a limb-viewing telescope with six spectral channels designed to image atmospheric $O_2$ airglow and noctilucent clouds. Although the main objective of the satellite mission is to observe structures in the airglow introduced by propagating smaller-scale waves, the airglow emissions are also subjected to large-scale dynamical disturbances, such as atmospheric tides and planetary waves. This work presents large-scale structures in the airglow field as observed by the MATS limb imager from February 2023 to April 2023. The ascending (north-going) node in the satellite orbit, corresponding to the local sunset, is dominated by a strong equator maximum in the dayglow. In contrast, the descending (south-going) node, corresponding to the local sunrise, indicates an accompanying equatorial minimum. These characteristics align with the expected behaviour of atmospheric tidal movements. Specifically, a downwelling of atomic oxygen is expected over the equator at local sunset, contributing to airglow chemistry and enhancing the emissions. Another distinct feature in the data is a westward propagating disturbance observed at high latitudes in the Northern Hemisphere, maximising in February, interpreted as the quasi-10-day planetary wave of zonal wavenumber 1.

## 1 Introduction

The MATS satellite mission (Gumbel et al., 2020) was launched from the Mahia Peninsula, New Zealand, in November 2022 to study the activity of gravity waves in the mesosphere and lower thermosphere (MLT). To identify atmospheric waves in the 70 - 110 km altitude range the satellite is equipped with a limb imager designed to observe wave perturbations introduced in noctilucent clouds (NLCs) and the global atmospheric airglow emissions. During winter and spring 22/23, MATS collected more than 4 million atmospheric limb images, distributed over six spectral regions. Through tomography and spectroscopy, these images will be used to retrieve a 3D representation of the MLT temperature field to fully characterise the gravity waves that perturb it.

Gravity waves (GWs) and planetary waves (PWs) originating in the lower atmosphere play crucial roles in the dynamics of the middle atmosphere. As air density rapidly decreases with altitude, the amplitudes of vertically propagating waves become substantial at MLT altitudes, and wave breaking results in momentum deposit in the form of wind motion. Atmospheric thermal tides also play a dominant role in shaping the atmospheric state of the MLT. The zonally uniform solar heating during





the day, together with the release of latent heat in tropical regions, gives rise to both migrating and nonmigrating tidal waves. These components move upward and influence the dynamics of the middle and upper atmospheres (Hagan and Forbes, 2002). In addition, the diurnal tide interacts with the PWs to generate nonmigrating tidal components (Lieberman et al., 2015), and dissipation of the tide substantially affects the zonal mean wind field (Hagan et al., 2009). Even without dissipation or breaking, these large-scale perturbations of the atmosphere constitute the background in which the smaller-scale waves propagate, present in the individual MATS images and in the retrieved 3D fields. To retrieve the wave properties of gravity waves, these large-scale structures must be separated from small-scale perturbations. The vertical flux of the horizontal gravity wave momentum (GWMF) is an important quantity for understanding the wave-driven changes of the MLT since it is directly related to the amount of deposited momentum. GWMF can be calculated directly from wave parameters derived from wave-induced perturbations of the temperature field (Ern et al., 2004). In this representation, the magnitude is determined by the ratio between the wave amplitude and the temperature of the medium in which the wave propagates. It is therefore important to separate the different scales with high precision.

The main purpose of this paper is to give a brief overview of the large-scale patterns observed by MATS, highlighting some of the most outstanding features within it, while at the same time providing a better understanding of the background field in which the smaller waves propagate. The study is based on the so-called Level 1b calibrated images (Megner et al., 2025), from which we have derived profiles of volume emission rate, as the MATS temperature retrieval is still under development.

## 2 MATS limb observations

The MATS satellite operates in a dawn-dusk, sun-synchronous orbit, at $\sim 585$ km altitude, collecting data from the MLT at roughly 17:30 and 05:30 local solar time (LST) at the equator. The main instrument of MATS is a six-channel (four IR + two UV) imager that targets the atmospheric limb, centred at a tangent altitude of approximately 90 km. MATS was launched on 4 November 2022 and completed its commissioning phase in January 2023. Further improvements were made during early 2023, and data collected from February resulted in three full months of consequent measurements. The satellite started experiencing issues with its attitude controls in May 2023 and scientific operations have been limited since then. A single day of MATS measurements includes a total of roughly 60,000 images taken by the four IR channels. For a given channel, an image of the atmospheric limb is captured every 6 seconds. The horizontal (across-track) distance in an image is approximately 200 km, and as images are taken at a fast rate, the same atmospheric volume is observed several times. This is exploited in the tomographic analysis to recover three-dimensional fields from the MLT. The limb imager measures atmospheric airglow in the $O_2$ atmospheric A-band, centred around 762 nm. The main channels IR1, (centred at 762 nm (3.5 nm bandwidth), and IR2, centred at 763 nm (8 nm bandwidth) capture the central and full width of the A-band, respectively, while the two background channels, IR3 and IR4, target spectral regions outside the band to quantify the contribution of Rayleigh scattered light in the images. This is required to isolate the signal solely attributed to airglow emissions.

In this study, we employ the MATS L1b data product, i.e. the calibrated images taken by the limb imager (Megner et al., 2025; Linder et al., 2025). Specifically, images from the IR2 channel are used to investigate global A-band emissions, as





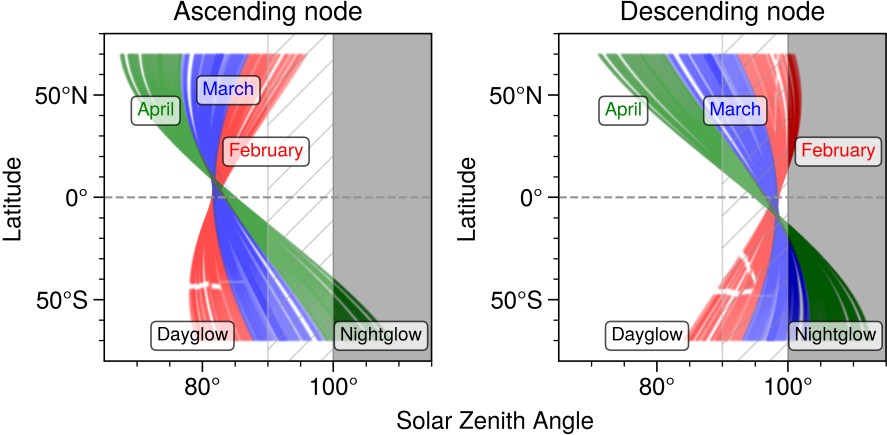

**Figure 1.** The geographical distribution of tangent point SZAs between $70°$S and $70°$S in the studied period.

this channel covers the entire A-band. The global structures are further analysed by performing a one-dimensional inversion, after subtracting the Rayleigh background using the background channel IR3. The measurements are known to be affected by

straylight, but it is believed that the error due to imperfect stray light compensation in the data used for this work is less than $5 \cdot 10^{13}$ photons $\mathrm{m}^{-2}\mathrm{sr}^{-1}\mathrm{s}^{-1}\mathrm{nm}^{-1}$ for the IR2 channel, i.e. less than 11 % of the mean peak radiance of all the MATS image columns used in this paper. As we are mainly interested in the large-scale variations connected to the peak of the emissions, straylight should not substantially impact this study. Straylight removal will be addressed in the ongoing MATS data processing development.

The $O_2$ A-band emission (centred around 762 nm) is separated into dayglow and nightglow, characterised by different chemical reactions, in turn determined by the solar conditions under which they occur. Dayglow emissions in the band are produced through reactions between $O(^1D)$ and $O_2$, where the short-lived $O(^1D)$ is generated by photolysis of $O_2$ or $O_3$. For nightglow conditions, emissions are produced through reactions initiated by the recombination of atomic oxygen in the $O(^3P)$ state, a longer-lived atomic oxygen species. The study of geographical and temporal changes in emissions is thus complicated

by the fact that these two types of emissions are not directly comparable, not least because the nightglow emissions are approximately an order of magnitude weaker than the dayglow emissions. A particularly strong dependence on the emission strength due to changes in SZA is observed for angles between $90°$ and $100°$. For readers seeking a more comprehensive introduction, we refer to Li et al. (2020).

The distribution of tangent point SZAs in the studied period, calculated at the central tangent height of the instrument, is

presented in Figure 1. The period is characterised by dayglow conditions during the ascending node ($\sim$ 17:30 LST at equator) while measurements in the descending node ($\sim$ 05:30 LST at equator) are mostly made in the transitional region between nightglow and dayglow. From the figure, it is clear that by avoiding SZAs $90°$ through $100°$, we restrict our study to focus mainly on dayglow emissions in the ascending node. Poles are ignored to avoid contamination by noctilucent clouds and auroras.





**Ascending node dayglow observations from IR2**

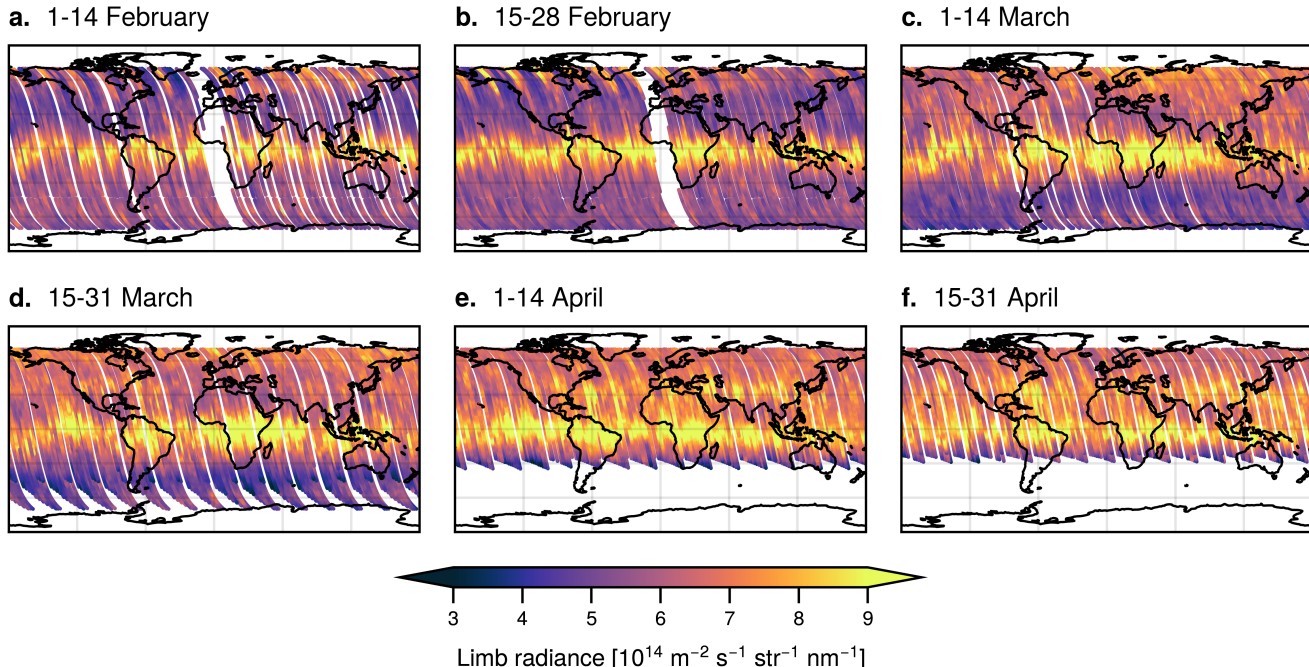

**Figure 2.** MATS ascending node IR2 measurements of limb radiance at 85 km tangent height made during February, March and April for $SZA < 93°$. Each point corresponds to an individual MATS measurement and every map of two weeks of observations. White stripes indicate gaps in the data. The data is collected at approximately 17:30 local solar time (LST) at the equator.

Global maps of the mean dayglow limb radiance at 85 km, observed at $\sim$ 17:30 LST (at the equator) by the IR2 channel, are shown in Figure 2 for February (2a, 2b), March (2c, 2d) and April (2e, 2f). Each panel consists of two weeks of individual measurements, so that two adjacent passes are separated by 24 hours with the longitude progressing by approximately 10° per orbit. The limb-radiance field is dominated by a large equatorial maximum, particularly prominent during February and March. The transition to nightglow for the ascending node is seen in the southern hemisphere during March and April, illustrated by the reduction in intensity. In April, a secondary peak at $\sim$ 30° becomes visible. The stripes over North America from 15 - 28 February indicate that over a large geographical region, a strong signal was registered during multiple orbits (each separated by 10° longitude).

Over the Pacific Ocean, the equatorial maximum deviates in part northward and southward from the equator, while remaining closer to 0°N over the Atlantic and African continents, a pattern that is consistent in the studied period.

Even in the regime of pure dayglow emissions, SZA is expected to have some effect on the measured limb radiance. Figure 3 examines how solar conditions affect the intensity of the limb radiance, depicting its variation with the SZA and the latitude for the ascending and descending nodes. As measurements in the descending node morning ($\sim$05:30 LST) are made in the





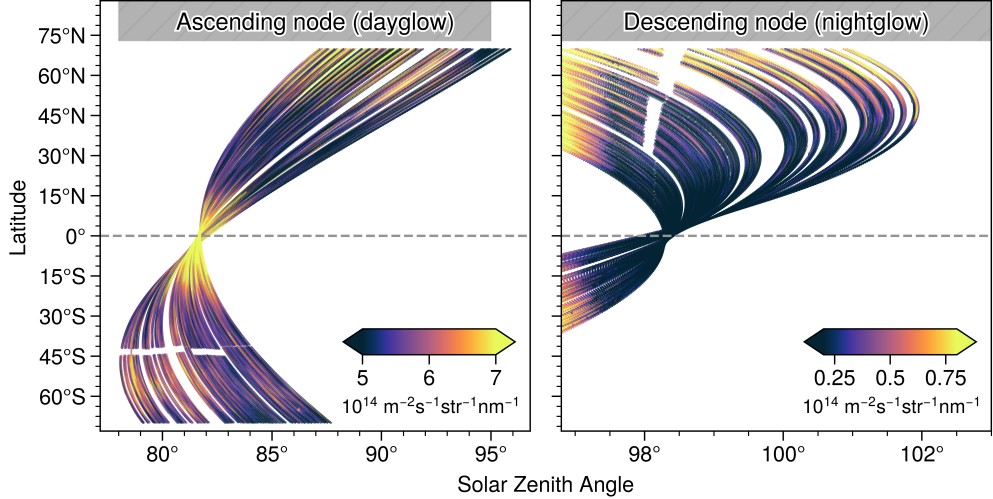

**Figure 3.** The mean limb radiance observed in the MATS IR2 at tangent height 85 km during February as a function of latitude and tangent point SZA. The left plot is obtained during evening dayglow conditions (∼17:30 LST at equator), while the right plot is obtained during morning nightglow conditions (∼05:30 LST at equator).

transition region between day- and nightglow, the visualisation of Figure 3 helps to distinguish how changes in SZA influence intensity separately from other geographical variations. From Figure 3 it is clear that the equatorial maximum in the dayglow of the ascending node and the equatorial minimum in the nightglow of the descending node exist independently of the lowest and highest SZA, respectively.

This paper focuses on large-scale airglow structures and only simple 1-D volume emission rate (VER) retrievals were performed. Tomographic 3-D data products aimed at studying small-scale phenomena, such as gravity waves, are still in development at the time of writing. A 1-D retrieval was performed for each IR1 and IR2 image. Each retrieval takes a single column of radiances (the mean of five central columns for each image was used) as input and returns the altitude profile of VER of the airglow in the corresponding spectral window. The forward model for retrieval assumes that the atmosphere is (locally) homogeneous and accounts for absorption by $O_2$. Emission and absorption are simulated using a line-by-line model with spectroscopic data from HITRAN (Gordon et al., 2022). These quantities have a significant dependence on temperature. For the purposes of this simple extraction, the temperature of MSIS climatology (Fomichev et al., 2002) was used. The main MATS Level 2 data products will include 3-D temperatures retrieved by combining IR1 and IR2 data. Inversions for 1-D retrievals were based on the classic Tikhonov regularisation scheme (e.g. Rodgers, 2000), and included the Euclidean norm and first-order Tikhonov operator-based regularisation terms.

Figure 4 presents several examples of the VER collected from tropical regions. The initial two passes are from 7 March, spaced about 6 hours apart, while the latter two passes are from 10 March and 11 March, respectively. The examples illustrate that the maximum signal occasionally occurs at latitudes lower than and higher than the equator and that the emission peak





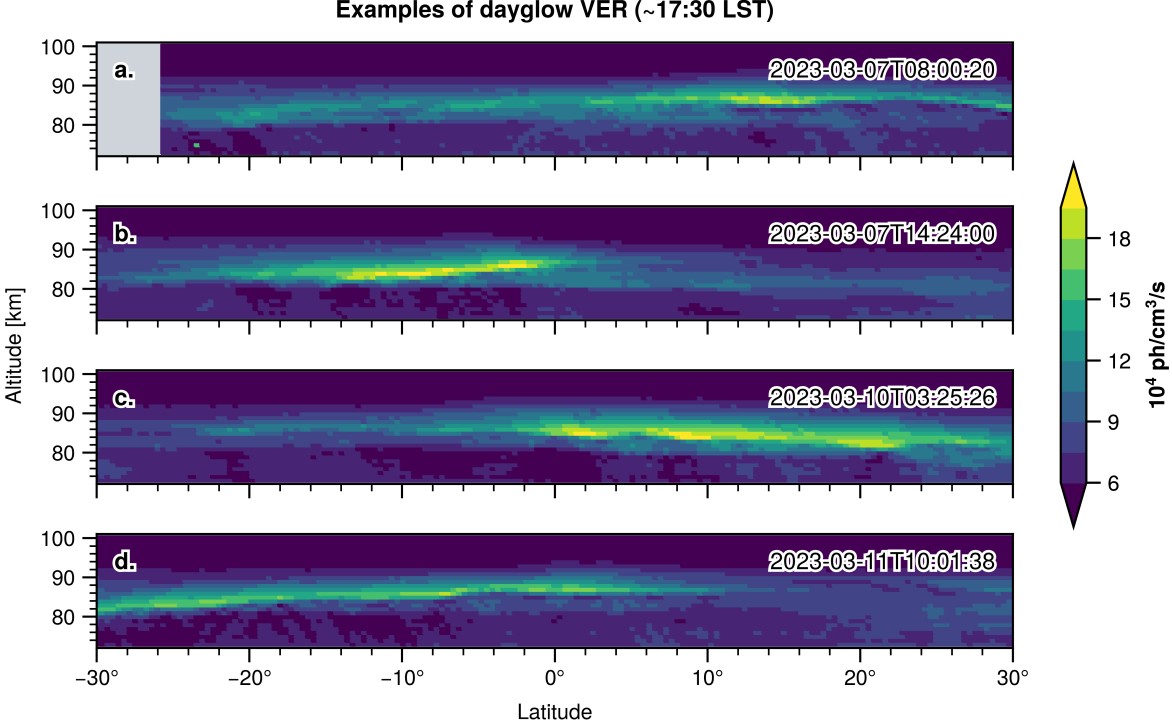

**Figure 4.** Four examples of VER latitude distributions taken from four different MATS passes over the tropics. The signals correspond to passes at approximately longitudes $140°$, $50°$, $215°$ and $115°$ (a-d order).

height varies slightly. The extent of the maxima also varies, as illustrated in Figures 4c and 4d, where the first example is characterised by strong and quite variable emission in the northern hemisphere, followed by an example with elongated signal enhancement in the southern hemisphere.

## 3 The equatorial maximum

The geographical extent of the VER observed at 85 km for February, March and April can be seen in Figure 5, where the mean of the VER is computed in $1° \times 1°$ longitude-latitude bins. For this emission height, the geographic location of the maximum is in good agreement with what was observed through the atmospheric limb (Figure 2). From Fig. 5, it is evident that the equatorial maximum still dominates the structure of global emission, with a mean signal remaining fairly constant throughout the studied period, approximately $15 \cdot 10^4$ ph cm$^{-3}$ s$^{-1}$, between $10°$S and $10°$N.

The oxygen A-band equatorial maximum has been observed in several other studies, for example, by HIRDI on the UARS satellite. Burrage et al. (1994) noted that the diurnal tide introduces a semiannual pattern in the equatorial O$_2$ nightglow, peaking around the equinoxes and diminishing around the solstices, speculating that vertical transport of atomic oxygen and

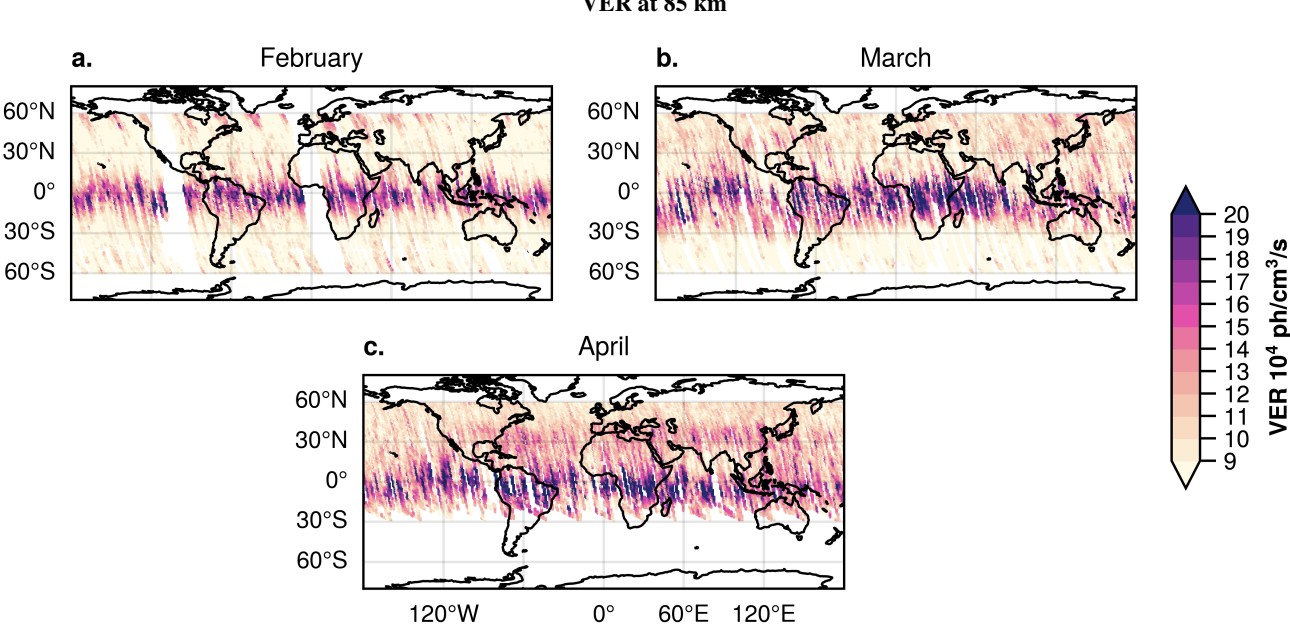

**Figure 5.** The mean A-band VER at an altitude of 85 km for February, March and April (∼ 17:30 LST at equator).

density changes play an important role. The mechanism for the increase in emissions was investigated by Ward (1998), who explored the dependence of local solar time by modelling vertical wind motions, emphasising that vertical wind advection

plays the most important role in global distributions of oxygen airglow, as changes in temperature and density perturbations should counteract the increase in emissions. Marsh et al. (1999) concluded that nightglow emissions peak in the afternoon due to changes in atomic oxygen recombination rates, while dayglow increases due to increased production of $O_3$, and therefore $O(^1D)$. To compare the airglow data measured by MATS with the predictions of tidal movements, a tidal model is required.

The Climatological Tidal Model of the Thermosphere (CTMT) serves as a tidal climatology for the MLT, derived from

observations made by SABER and TIDI on TIMED from 2002 to 2008 (Oberheide et al., 2011). The model includes the largest tidal components, both semidiurnal and diurnal. Superposing six diurnal components and eight semidiurnal components, the longitudinal and latitudinal distributions of the tidal perturbations are isolated for an LST of 17:30 and an LST of 05:30 at the equator. In Figure 6 the expected vertical wind perturbations are shown for these two local times for February, March, and April, at an altitude of 85 km. The evening wind field is characterised by a strong downdraft at the equator and updrafts at

mid-latitudes. The situation is reversed in the morning with an updraft at the equator and a downdraft at the mid-latitudes. The wind patterns are symmetric around the equator apart from a northward stretch over the Pacific Ocean and a slight southward extension over South America, similar to what is observed in the MATS VER field. An expected peak at the equinox would imply that March and April would be of similar strength, while February would be weaker. This is indicated by the wind field in Figure 6.





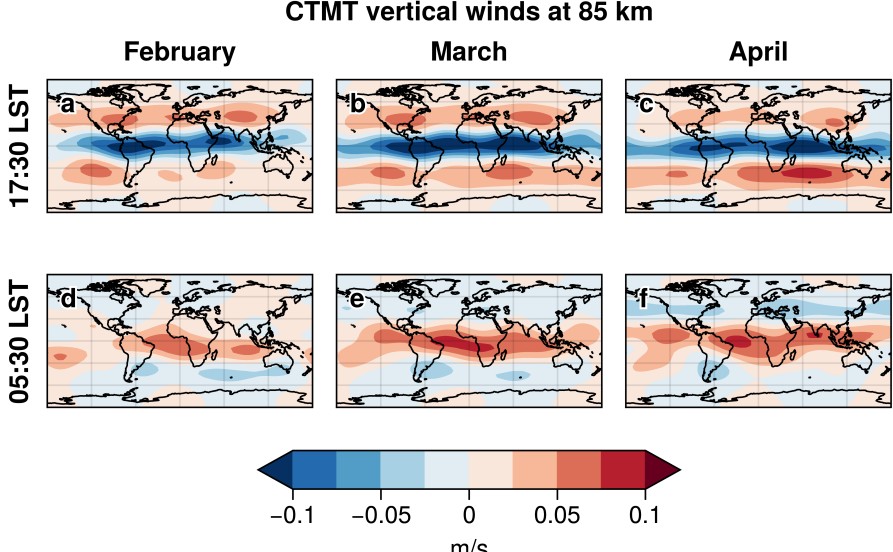

**Figure 6.** Vertical wind fields from the CTMT model at 85 km for 17:30 LST (top) and 05:30 LST (bottom).

Figure 7 allows for an analysis of the changes in the height of the emission peak, the intensity of the emission, and their longitudinal distribution. The average equatorial VER is determined between $10°$ S and $10°$ N for each of the three months. The figure shows that the signal is consistently strong over the African continent, between $0°$ and $50°$, but from $300°$ to $50°$, the peak emission is stronger during March and April and weaker in February. In addition, in February, the emissions in these regions also occur at lower altitudes, particularly noticeable around $340°$ and $30°$. Observed longitudinal variations are

expected from the nonmigrating tidal components and the vertical advection they introduce. According to CTMT wind data, around the end of February, a rise in VER is anticipated over the Indian Ocean ($50°$ - $100°$) due to increased downwelling. This is confirmed in Fig. 7. However, no increase in signal is observed over Southeast Asia, in contrast to the CTMT climatology. Obviously, given that the model is based on 6-year averages and the MATS derives from a single-year's measurements, a detailed direct comparison is of limited relevance.

## 4    A propagating high latitude disturbance

In Figure 2, in the limb radiances from February, short periods of high VER are seen at high northern latitudes (apparent as yellow stripes). By comparing Figs. 2a and 2b, it is observed that these perturbations first occur most prominently over Asia, to then dominate the measurements over the Americas. In the latter half of February, over North America, several MATS passes recorded low emissions across the 50°N-70°N latitudes. Measurements of drastically increased emissions in the same

region then followed these passes. These enhancements indicate a periodic change in emission, characteristic of planetary wave activity. To further study this, the VER data between 60 °N and 70 °N are organised into 20° longitude bins and are presented

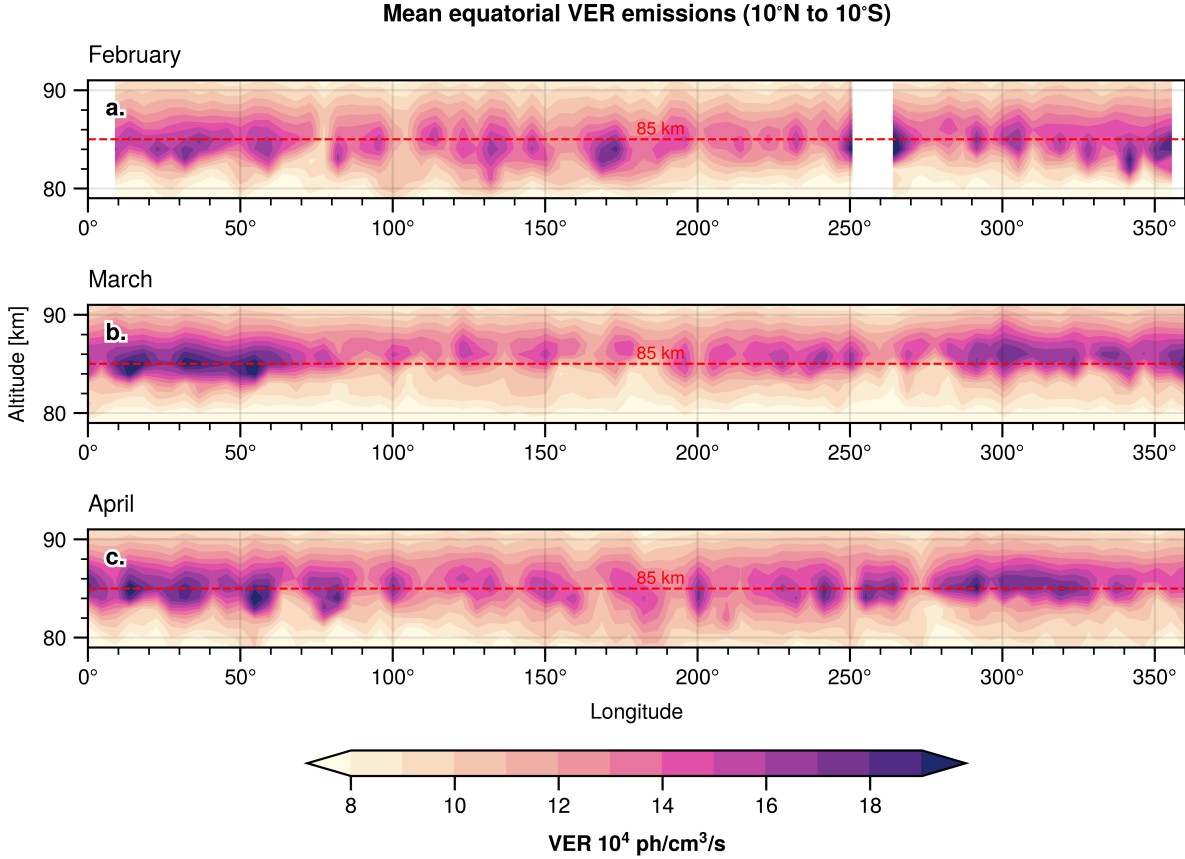

**Figure 7.** Oxygen A-band VER in the equatorial region over February, March and April.

as a Hovmöller diagram in Figure 8. The figure evidently shows a significant large-scale emission peak migrating westward throughout February. The temporal separation between the maxima of approximately 10 days suggests that the perturbation may be due to the quasi–10-day wave with zonal wave number 1 (Q10DW). This planetary wave has been studied, for example, using TIMED/SABER temperature measurements obtained over 2002 - 2013 by Forbes and Zhang (2015). The study shows that Q10DW is strongest during winter around 50°N with typical temperature amplitudes of around 5K at 100 km. Furthermore, the phase of the wave is approximately constant with height and thus propagates horizontally but hardly in the vertical direction. The study is limited by the latitudinal extent, which only reaches ±50°, but the wave has an expected (theoretical) maximum at 55°. In the same study, the amplitude peaks between 80-100 km. The temperature amplitude is thus similar to the corresponding one seen in the CTMT fields at this altitude, which is ∼ 6K at the equator.

During the 2018 sudden stratospheric warming (SSW), the influence of warming on Q10DW in the MLT was investigated up to a latitude of 53.5° N using meteor radars. The study illustrated that the SSW amplified the wave amplitude (Luo et al., 2021). In 2023, a smaller SSW occurred during the end of January followed by a major SSW in the middle of February (Vargin

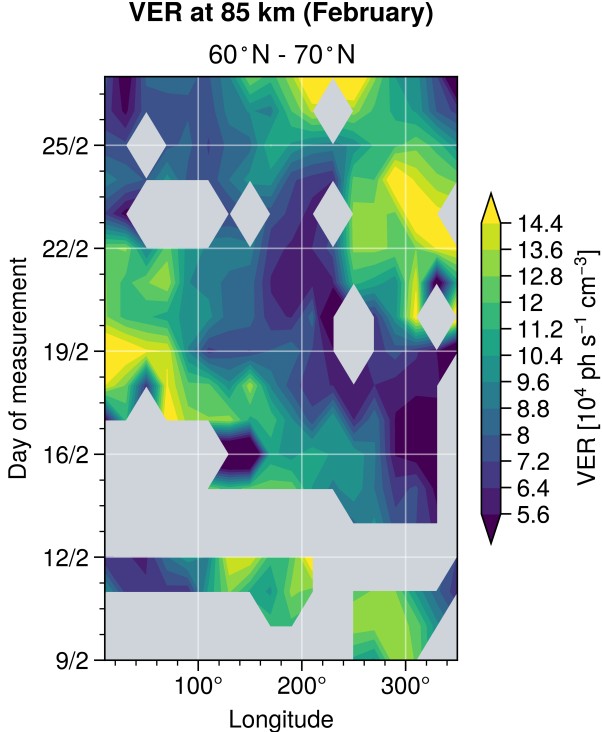

**Figure 8.** A Hovmöller diagram illustrating the high latitude VER, grouped into 20° longitude bins, observed at an altitude of 85 km during February

et al., 2024). This may have enhanced the effect that the Q10DW had on the airglow emissions and possibly explains why this
propagating planetary wave becomes a dominating feature in the MATS measurements.

## 5  Conclusions

We here present the first global results of the MATS airglow observations. We have found that tidal dynamics dominate the
variations in the airglow emission field observed by MATS around the spring equinox. This is revealed by a strong equatorial
maximum in the dayglow around local sunset, accompanied by an equatorial minimum at local sunrise. The structure has
significant longitudinal and latitudinal variations during this short 3-month period, and tendencies of a seasonal cycle are
observed. The observed pattern is in agreement with observations by previous satellite missions that studied oxygen airglow
emissions. Because MATS is in a sun-synchronous orbit, tides and their influence on the dynamics are recorded at just two
specific local times, and possibilities to study the general behaviour of the tide are therefore limited. However, temporal and
longitudinal variations at these two specific phases can be analysed with a high and consistent sampling rate. Tidal wind
patterns will consistently affect the observed wave field in the collected data in the two phases. Horizontal wind patterns





introduce filtering effects that may need to be taken into account if a climatology of MLT waves is to be derived. More directly, tides can affect measurement analysis by controlling the intensity of the airglow. A minimum in nightglow emissions recorded over the equator could have implications for how well measurements can be performed, and eventually how well temperatures can be derived.


*Code and data availability.* The analysis scripts are available on request. The MATS level 1b dataset can be accessed from the Bolin centre database at https://doi.org/10.17043/mats-level-1b-limb-cropd-1.0. The code used to produce the level 1a and level 1b datasets is available via the GitHub repositories https://github.com/innosat-mats/level1a and https://github.com/innosat-mats/MATS-L1-processing, respectively.


*Author contributions.* BL: formal analysis, writing, visualisation; LK: performed and introduced the retrieval of volume emission rate; LM
& DM: conceptualisation and discussions.

*Competing interests.* The authors declare they have no competing interests.





*Acknowledgements.* B. Linder, D. Murtagh, L. Krasauskas and L. Megner received funding from the Swedish National Space Agency (grant nos. 2012-01684, 210/19, 2021-00052, and 2022-00108).



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
