# Peer review of "The global $O_2$ airglow field as seen by the MATS satellite: strong equatorial maximum and planetary wave influence"

_EGUsphere, 2025_

## Author Comment (AC1)

We thank Anonymous Referee #1 for the valuable comments and for the suggestions on future work. Please find our answer to the comments below. Answer given as green text.
* * *
*This paper reports on the nature of large-scale structures observed in Feb-Apr 2023 O2 limb emissions from the MATS satellite, which is in a Sun-synchronous ascending/descending dusk/dawn (17:30/05:30) orbit. It is well-written, provides new insights into the interaction between dynamics and chemistry, and is a nice addition to the literature.*

specific comments

The authors do an excellent job explaining the complexities of the measurements in terms of dependencies on ascending vs descending, dayglow vs. nightglow, solar zenith angle, and month, so that geographical dependencies in the emissions can be ascribed to the underlying atmospheric dynamics.

The authors make creative use of the CTMT vertical winds to explain the equatorial enhancements in the 17:30 LST emissions at the equator and some aspects of the latitude and longitude dependencies, although this approach is limited by the fact that the Feb-April 2023 dynamics probably differs to any unknown degree from the 7-year climatology represented by the CTMT.

The authors also credibly argue that the emissions at high latitudes are modulated by the quasi-10-day planetary wave.

The arguments put forth by the authors in the context of previously published work is at about the right level, and the references to the literature are sufficient.

In future work the authors may wish to consider looking for eastward-propagating 2d-4d period ultra-fast Kelvin wave signals in their 10S-10N VER emissions.

I recommend publication of this paper as is.

technical corrections

Figure 1 caption should explain the meaning of the grey shading and cross-hatching.

We agree that an explanation for the shading and cross-hatching is missing. The caption and the text referring to the figure has been updated in the new version of the manuscript.

---

## Author Comment (AC2)

We thank Anonymous Referee #2 for the valuable comments. Please find our answers to the comments below. The main comment is answered first, and the other comments one at a time. Answers are given in green / blue.
* * *
*The paper shows a first geophysical interpretation of measurements by the MATS satellite. A new satellite instrument is a major effort and confidence to the data needs to be gained and spread to the community by checking that such geophysical interpretations make sense. Therefore the study merits publication, even though not every detail is understood. However, the presented graphics should be made more consistent and I would request that easier comparison is facilitated.*

Main comment:

**The figures need more consistency: Give the main latitude lines at the left of the maps (once for each column) and the main longitudes once per each row. Use the same color for the same things e.g. (Volume emission rates currently jumping between viridis and a second color scale). Figure 7: Use the same longitude axis as for the maps -180° to 180°. I think it would be good to plot the vertical winds in the same format here as well.**

**We agree with the main comment – the figures should be more consistent to simplify readability and comparison. We have updated the figures as suggested, along with some minor adjustments:**

**Units: In a specific comment below, it was noted that photons were missing in the units of Figure 2. We have now included photons in the units of Figure 2 and Figure 3. Additionally, the VER units of Figure 5, Figure 7 and Figure 8 are now written in the same format to simplify readability.**

**Maps: The main longitude and latitude lines are now included with the maps. Figure 2 has been updated as well as Figure 5 and Figure 6. We interpret the comment above to mean that longitude labels should be once per column, and latitude labels once per row (the opposite was suggested). In the latter two figures, more space allowed for labels in several subplots.**

**Colormaps: We agree that it would be good to have common colormaps for the same parameter. We have updated Figure 4 and Figure 8 (that show VER) so that they no longer have 'viridis' but cmap 'sunset' for consistency between all VER plots.**

**Latitude/longitude ranges: Figure 7 and Figure 8 are now updated to longitude ranges from 180W to 180E, and for full consistency Figure 4 is now labeled as 30S to 30N instead of -30 to 30. The manuscript text referring to longitude and latitude has been revisited and updated.**

**Vertical wind: The vertical winds from CTMT are plotted in Figure 7 for easier analysis, as was suggested. The text has been updated accordingly.**

Suggestions:

**a) For F7 Maybe have a look at JAWARA as well. What longitude structures do they have in their vertical winds?**

This is a very good idea and we have considered using JAWARA. However, this is out of scope for the current manuscript as it aims to give a general overview and highlight features that will be further analysed in coming studies. See next comment.

**b) For F8 Does SABER cover these latitudes? Could you do a similar Hovmoeller plot?**

We agree that a comparison with SABER could be a next step, but this also means comparison with a higher level data product – temperature. The oxygen emission is complex, it depends on a multiple of parameters such as temperature, availability of constituents and density. When MATS temperature product is finalised deeper analysis along the suggested lines becomes possible.

Specific comments

**L30 sentence structure?**

We agree that the sentence was confusing. The sentence has been adjusted in the updated manuscript.

**L35 compare to completely ignoring PWs and tides, yes. But really more improtant is that GW amplitudes are usually smaller than these of Rossby waves and tides, i.e. that you get wrong GW amplitudes by wrong attribution in the fit. By subtracting the global scale waves first, you can also determine GWs larger than the analysis volume**

When we write 'the medium in which the wave propagates' we mean the ambient atmosphere including large-scale perturbations such as planetary waves and tides. This is now elaborated in the updated manuscript.

**L55 required to separate airglow emission signal from Rayleigh scattered background ?**

This is correct. We now specify that IR3 and IR4 can be used to remove the Rayleigh contribution in the images taken by IR1 and IR2.

**L69 species   -> would mean for me rather different molecules not states of the same molecule**

We agree, we have changed "species" to "state".

**F2 is this photons? photon counts? All real units are in the denominator  ;**
 **... and every map to two weeks …**

The real units are indeed photon counts, and we have updated the units to include "ph". As was mentioned in the main comment, Figure 3 now has photons included in the units as well.

Upon close inspection of the plots shows that including more than 14 days in the plot does generate some overlapping points, and as is suggested, the plots are now divided into days 1-14 and 15-28. This has no effect on the discussion and results.

**L91 ... examines how ... In the context I was first assuming that you would do a model calculation to separate effects, Reformulate slightly to make evident that this are your measurements.**

A good suggestion, we have rewritten the sentence for clarity.

**L97 Motivation was given above, just state: For this study 1-D …**

This would be an improvement. We have updated the manuscript with the suggestion.

**L131 Superposing six ... perturbations are isolated …**
**Why isolate? What did you do here?**

The phrasing is misleading, they are not isolated but rather "shown". Corrected in the updated manuscript.

**F7 Maps run from 180W to 180E -> Do the same here! You could include three panels for the tidal winds**

The map has now been updated to go from 180W to 180E. Additionally, the wind is now included in the same plots, which makes comparison much easier. The text has been adjusted to match the new figure.

**L149 does the model provide sigma? If yes you could check whether this is inside natural variations of the previous period**

The model does not seem to provide any sigma. However, based on a study by Hays et al. 2003, it is clear that the longitudinal distribution of the oxygen airglow tidal signature changes on short timescales – even though the zonal mean is relatively constant. It is therefore natural to assume that the CTMT climatological mean could differ from the MATS measurements. Reference at the end of this document.

**L176 in general agreement - you certainly have not shown to be compliant in details**

We agree that the agreement is general. Updated.

**L182 do you really want to conclude your paper this way, by an unquantified speculation about shortcomings in your temperature retrievals. In general, I would change the motivation slightly to general performance.**

The suggestion is good. We have updated the conclusion to focus more on the results and we have added a short outlook of what can be done once the final MATS temperature product has been retrieved.

**References**

Hays, P. B., J. F. Kafkalidis, W. R. Skinner, and R. G. Roble (2003), A global view of the molecular oxygen night airglow, *J. Geophys. Res.*, 108, 4646, doi:10.1029/2003JD003400, D20.